# A Fast Bistatic ISAR Imaging Approach for Rapidly Spinning Targets via Exploiting SAR Technique

**Zhijun Yang [1,2], Dong Li [2], Xiaoheng Tan [2,*], Hongqing Liu [3] and Guisheng Liao [4]**

[1]  State Key Laboratory of Power Transmission Equipment & System Security and New Technology, Chongqing University, Chongqing 400044, China; yangzj158@cqu.edu.cn

[2]  Chongqing Key Laboratory of Space Information Network and Intelligent Information Fusion, School of Microelectronics and Communication Engineering, Chongqing University, Chongqing 400044, China; lid0705@cqu.edu.cn

[3]  Chongqing Key Laboratory of Communications Technology, Chongqing University of Posts and Telecommunications, Chongqing 400065, China; hongqingliu@cqupt.edu.cn

[4]  National Laboratory of Radar Signal Processing, Xidian University, Xi'an 710071, China; liaogs@xidian.edu.cn

*  Correspondence: txh@cqu.edu.cn; Tel.: +86-158-7045-9537

**Abstract:** Because of the large range of cell migration (RCM) and nonstationary Doppler frequency modulation (DFM) produced by non-cooperative targets with rapid spinning motions, it is difficult to efficiently generate a well-focused bistatic inverse synthetic aperture radar (ISAR) by use of the conventional imaging algorithms. Utilizing the property of the inherent azimuth spatial invariance in strip-map synthetic aperture radar (SAR) imaging mode, in this work, an efficient bistatic ISAR imaging approach based on circular shift operation in the range-Doppler (RD) domain is proposed. First, echoes of rapidly spinning targets are transformed into the RD domain, whose exact analytical is derived on the basis of the principle of stationary phase (POSP). Second, the RCM is corrected by using an efficient circular shift operation in the RD domain. By doing so, the energies of a scatterer that span multiple range cells are concentrated into the same range cell, and the time-varying DFM can also be compensated along the rotating radius direction. Compared with existing methods, the proposed method has advantages in its computational complexity, avoiding the interpolation and multi-dimensional search operation, and in its satisfactory imaging performance under low signal to noise ratio (SNR) conditions thanks to the two-dimensional coherent integration gain utilized. Finally, several numerical simulations are conducted to show the validity of the proposed algorithm.

**Keywords:** bistatic inverse synthetic aperture radar; range-Doppler domain; rapidly spinning targets; circular shifting operation

## 1. Introduction

The inverse synthetic aperture radar (ISAR) plays a crucial role in the detection, recognition or identification of rapidly spinning targets [1–3], because electromagnetic images of non-cooperative targets with rapidly rotating motions can be obtained. Conventional studies of ISAR imaging have focused mainly on monostatic radar configurations in which a transmitter and a receiver are collocated. Nevertheless, the monostatic ISAR technique suffers from restrictions such as imaging problems for stealthy targets. While detecting and utilizing a monostatic radar configuration, the electromagnetic energy is reflected by the stealthy targets in the direction other than that of the receiver line of sight (LOS), which yields a decrease in the signal to noise ratio (SNR) for a received radar echo signal [4–6].

Compared with monostatic ISAR, the bistatic radar configuration in which the transmitter and receiver are spatially separated presents advantages in radar detection scope, concealment,

anti-interference capability and the SNR of the received echo. In [7], the geometric structure of a bistatic ISAR with fixed transmitter and receiver configurations is set up, and the range resolution and azimuth resolution are analyzed. The applicability of employing monostatic ISAR techniques to constitute bistatic configuration is analytically demonstrated and the point spread function (PSF) of a bistatic ISAR system is derived in [8]. At the same time, the influence of distortion and defocusing for bistatic ISAR images caused by the nonstationary bistatic angle is also discussed. In [9], the robustness of distortion and defocusing for bistatic ISAR images with phase synchronization errors is introduced. The distortion term and defocusing term via expending the bistatic angle, utilizing first-order Taylor expansion, are analyzed in [10]. The distortion term on the basis of the image plane and synthetic vector is derived in [11]. However, the connection between the distortion term and the time-varying bistatic angle is not presented. The distortion term is compensated via using a novel algorithm in [12], but this method is invalid for non-cooperative imaging targets. In [13], a method is presented to achieve bistatic ISAR image distortion correction by using the coefficients of polynomial phase signal in the range bin that contains a prominent scatterer. However, this method is seriously dependent on the prominent scatterer, which restricts its application in a real situation. Then, on the basis of the particle swarm optimization (PSO), the parameter estimation technique is studied in [14], in which the phase history data is extracted from the prominent scatterer. However, it also relies on the prominent scatterer and the computational complexity is high. In [15], the Doppler migration caused by the geometric distortion of the bistatic configuration and the target motion is compensated by constructing the phase shift function with the estimated shift factor. After that, a specific geometric characteristic of the target is used to correct the bistatic distortion. However, the geometric characteristic for non-cooperative targets is usually not easy to acquire. In [16], the bistatic ISAR image is reconstructed, but it only considers the near-field targets. The range of cell migration (RCM) correction approach for a well-focused synthetic aperture radar (SAR) is proposed in [17], which is based on the estimation for the dynamic parameters of targets, which is introduced in [18]. Furthermore, the RCM correction algorithms based on Keystone transform are presented in [19,20]. Although, in the case of small rotation angles, those methods can produce well-focused ISAR images for maneuvering targets, the real-time performance and effectiveness of those methods are insufficient to obtain images for targets with rapid spinning motions because the large rotation angle of scatterers generates large RCM and nonstationary Doppler frequency modulation (DFM). In the real world, ISAR imaging must use signal processing methods to characterize signals of interest [21,22].

The micro-Doppler features of targets are subject to Doppler modulations that provide useful information for targets to extract dynamic properties. In [23], the motion characteristics of objects are exploited by using the time-varying Doppler characteristics. Furthermore, the method of presenting the Doppler signature of objects with micro-motions is interpreted [24]. The micro-Doppler features are regarded as a new and unique radar signature of targets that can be utilized to distinguish the micro-Doppler feature of different micro-motions. Based on the micro-motions between warheads and decoys, the micro-Doppler feature can be extracted for target recognition. The precession angle of midcourse targets is estimated on the basis of the vibration for two scattering centers' projection distances on a high-resolution range profile (HRRR) sequence in [25]. The gait characteristic is extracted from the micro-Doppler features [26].

To consider the large RCM and nonstationary DFM of targets in the case of rapidly spinning motion, motived by the property of azimuth spatial invariance, a fast bistatic ISAR imaging method to target rapid spinning motions via exploiting the SAR technique is proposed in this work. To that end, first, the echoes of rapidly spinning targets are transformed into the RD domain, and the accurate analytical derivation of them is derived based on the principle of stationary phase (POSP) simultaneously. Second, by utilizing an efficient circular shifting operation in the range-Doppler domain, the large RCM is corrected, and the time-varying DFM can also be compensated along the rotating radius direction. Finally, the well-focused bistatic ISAR images are generated. Thanks to the absence of interpolation and multi-dimensional search operations, the computational complexity of

the presented approach decreases dramatically. In addition, the imaging performance of our proposed method is superior under low SNR environments because of the two-dimensional coherent integration gain utilized.

The remainder of this paper is organized as follows. In Section 2, we establish the geometry and signal model for bistatic ISAR imaging. Thus, the bistatic ISAR imaging method is developed. In Section 3, several imaging results and analyses based on simulated data are introduced. Some conclusions are provided in Section 4.

## 2. Bistatic ISAR Imaging for Targets with Rapidly Spinning Motion

### 2.1. Geometry and Signal Model

In this section, the introductions and assumptions about geometry and signal models are present to facilitate the following discussion. Firstly, it is assumed that the translational motion compensation (TMC) is conducted [27,28]. Secondly, the rotation axis $z = i_{\text{eff}}$ and rotation angular velocity $\Omega(t_a)$ remain stationary in the imaging interval, where $t_a$ denotes slow time. Thirdly, the bistatic angle $\beta(t_a)$ remains constant under the far-field condition and can be accurately estimated by solving the triangle that is composed of the target and bistatic radars.

Figure 1 shows the geometry model for bistatic radar configuration, the rotating center is the origin $O$, $T_X$, $R_X$, $R_{TX}(t_a)$ and $R_{RX}(t_a)$ denote the transmitter, receiver, the distances from $T_x$ and $R_x$ to rotating center $O$, respectively. A Cartesian coordinate system $(x, y)$ is constructed to explain the bistatic ISAR model, where the $x$ axis is aligned with $i_{\text{BI}}(0)$, as shown in Figure 1.

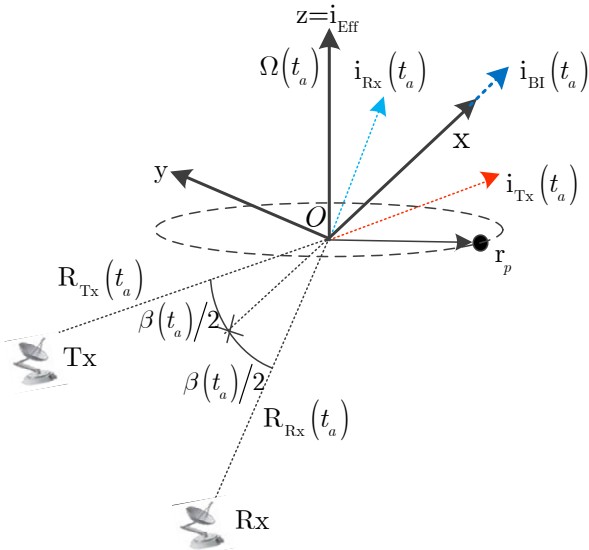

**Figure 1.** Geometry model of bistatic inverse synthetic aperture radar (ISAR) imaging.

Assuming that the radar transmits linear frequency-modulated (LFM) signals with bandwidth $B_r$ and wavelength $\lambda$, the received signal from scatterer $P$ at $r_P = (r_p, \varnothing_p)$ after demodulation and range compression is given by

$$ss(r, t_a) = \sigma_p w(t_a) \text{sinc}\left[\frac{\pi B_r}{c}(r - R_P(t_a))\right] exp\left\{-j\frac{2\pi R_p(t_a)}{\lambda}\right\} \tag{1}$$

where $\sigma_p$, $w(t_a)$, and $c$ represent the scattering coefficient, azimuth modulation function of the antenna, and the speed of light, respectively. In (1), $R_P(t_a)$ denotes the instantaneous slant range of scatterer $P$, and it is

$$R_P(t_a) = R_{\text{BI}}(0) + 2\cos(\beta(0)/2)r_p \sin(\omega t_a + \varnothing_p) \tag{2}$$

where $R_{BI}(0) = R_{TX}(0) + R_{RX}(0)$.

From (1) and (2), the instantaneous Doppler frequency of the rapidly spinning target can be written as

$$f_d(t_a) = -\frac{1}{2\pi}\frac{d}{dt_a}\left[-\frac{2\pi R_p(t_a)}{\lambda}\right] = \frac{2\cos(\beta/2)r_p\omega\cos(\omega t_a + \phi_p)}{\lambda} \tag{3}$$

Based on the analytical expression of instantaneous Doppler frequency $f_d(t_a)$, the Doppler bandwidth $B_d$ of the target is given by

$$B_d = \frac{4\cos(\beta/2)r_p\omega}{\lambda} \tag{4}$$

From Equations (1)–(4), compared with maneuvering targets, the instantaneous slant range $R_p$ contains a trigonometric function that will result in a large RCM and non-stationary DFM. Therefore, the well-focused bistatic ISAR image cannot be obtained by utilizing conventional imaging approaches. The diagram of range compression is manifested in Figure 2a, where two red lines denote the envelopes of two scatterers with the same rotating radius and different initial phases, and the blue line denotes another with a different rotating radius and initial phase.

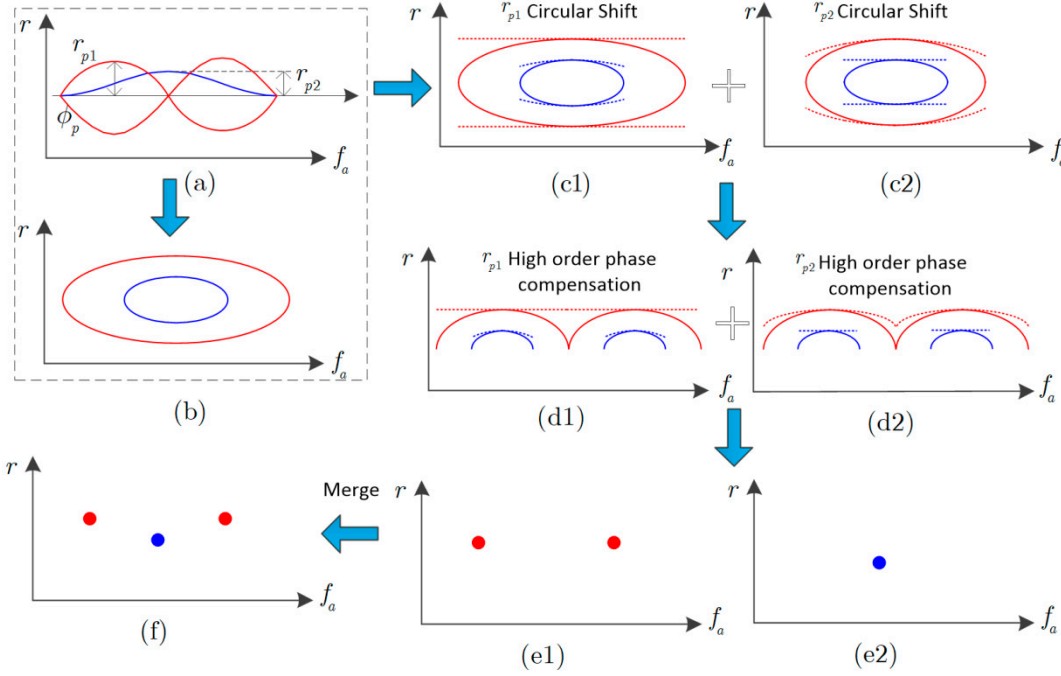

**Figure 2.** Bistatic ISAR imaging procedure illustration for proposed method.

### 2.2. Proposed Imaging Method

It is well known that the generalized radon transform (GRT) [29,30] method and Hough transform (HT) [31] are developed to perform ISAR imaging for space non-cooperative targets with rapid spinning motions under the conditions of large RCM and non-stationary DFM. For GRT and HT approaches, the energies of the scatterers are non-coherently integrated along the sinusoidal envelope calculated via rotating radius $r_p$ and initial phase $\varnothing_p$ after range compression, as shown in Figure 2a. Therefore, the sole peak is obtained while the search trajectory is matched with the true one. However, the disadvantage of those methods is that the computational complexity is burdened due to the multi-dimensional search operation. Furthermore, the SNR gain is low since non-coherent accumulation is used. Therefore, a new approach that is lightweight and robust to the noise should be further developed for spinning targets.

To effectively obtain bistatic ISAR images for space non-cooperative targets with rapid spinning motions, in this work, a fast approach based on the circular shift operation is presented to correct

the large RCM and compensate for nonstationary DFM via exploiting the SAR technique [32]. The advantage is that the computational efficiency is improved because the RCM of the echoes for multiple scatterers with the same rotating radii are corrected for once, and the diagrams are provided in Figure 2a,b, and the necessary derivations are given as follows, by conducting transform (FT) along $t_a$ in Equation (1), given by

$$sS(r, f_a) = \int \sigma_p w(t_a) \cdot sinc\left[\frac{\pi B_r}{c}(r - R_P(t_a))\right] exp\left\{-j\frac{2\pi R_p(t_a)}{\lambda}\right\} exp(-j2\pi f_a t_a) dt_a \tag{5}$$

It should be pointed out that the integral result directly obtained from Equation (5) is difficult. However, it can be determined via the stationary point of the phase term in Equation (5) on the basis of the POSP. Therefore, the stationary point is obtained by taking the derivative of the phase term $\partial(t_a) = -\frac{2\pi R_p(t_a)}{\lambda} - 2\pi f_a t_a$ in regard to $t_a$ and setting it to be zero yields.

$$\frac{d[\partial(t_a)]}{dt_a} = -\frac{2\pi\left[2\cos(\beta/2)r_p\omega\cos(\omega t_a + \varnothing_p)\right]}{\lambda} - 2\pi f_a = 0 \tag{6}$$

Solving Equation (6) produces two stationary points $t_a^*$ as

$$t_a^* = \begin{cases} -\frac{1}{\omega}\left\{\sin^{-1}\left(\sqrt{1 - \left(\frac{f_a\lambda}{2\cos(\beta/2)r_p\omega}\right)^2}\right)\right\}, & (\omega t_a + \phi_p) \in [-\pi, 0); \\ \frac{1}{\omega}\left\{\sin^{-1}\left(\sqrt{1 - \left(\frac{f_a\lambda}{2\cos(\beta/2)r_p\omega}\right)^2}\right)\right\}, & (\omega t_a + \phi_p) \in [0, \pi]; \end{cases} \tag{7}$$

Substituting Equation (7) into Equation (5), the analytical expression in RD domain is formulated as

$$sS(r, f_a) = \begin{cases} \sigma_p w(f_a/B_a) \cdot sinc\left[\frac{\pi B_r}{c}\left(r - \left(R_{BI}(0) - 2\cos(\beta/2)r_p\sqrt{1 - \left(\frac{f_a\lambda}{2\cos(\beta/2)r_p\omega}\right)^2}\right)\right)\right] \\ \times exp\left(j2\pi f_a\frac{\varnothing_p}{\omega}\right)exp(j\varphi_1(r_p, f_a)), (\omega t_a + \varnothing_p) \in [-\pi, 0); \\ \sigma_p w(f_a/B_a) \cdot sinc\left[\frac{\pi B_r}{c}\left(r - \left(R_{BI}(0) + 2\cos(\beta/2)r_p\sqrt{1 - \left(\frac{f_a\lambda}{2\cos(\beta/2)r_p\omega}\right)^2}\right)\right)\right] \\ \times exp\left(j2\pi f_a\frac{\varnothing_p}{\omega}\right)exp(j\varphi_2(r_p, f_a)), (\omega t_a + \varnothing_p) \in [0, \pi]; \end{cases} \tag{8}$$

where $w(f_a/B_a)$ is the Doppler bandwidth window, $\varphi_1(r_p, f_a)$ and $\varphi_2(r_p, f_a)$ are the high order phase terms, respectively, and they are

$$\varphi_1(r_p, f_a) = -\frac{4\pi}{\lambda}\left(R_{BI}(0) + 2\cos(\beta/2)r_p\sqrt{1 - \left(\frac{f_a\lambda}{2\cos(\beta/2)r_p\omega}\right)^2}\right) \\ -2\pi f_a\frac{1}{\omega}\sin^{-1}\left(\sqrt{1 - \left(\frac{f_a\lambda}{2\cos(\beta/2)r_p\omega}\right)^2}\right) \tag{9}$$

$$\varphi_1(r_p, f_a) = \frac{4\pi}{\lambda}\left(R_{BI}(0) + 2\cos(\beta/2)r_p\sqrt{1 - \left(\frac{f_a\lambda}{2\cos(\beta/2)r_p\omega}\right)^2}\right) \\ +2\pi f_a\frac{1}{\omega}\sin^{-1}\left(\sqrt{1 - \left(\frac{f_a\lambda}{2\cos(\beta/2)r_p\omega}\right)^2}\right) \tag{10}$$

From Equation (8), what is noteworthy is that the initial phase $\varnothing_p$ is separated from the high order phase term and the sinc envelope. Thus, computational efficiency is improved because the initial phase $\varnothing_p$ has not been considered. The energy trajectory of scatterers in the RD domain is determined by

$$\mathcal{R}\left(r_p, f_a\right) = \begin{cases} r - \left(R_{BI}(0) - 2\cos(\beta/2)r_p\sqrt{1 - \left(\frac{f_a\lambda}{2\cos(\beta/2)r_p\omega}\right)^2}\right), \\ \qquad\qquad \left(\omega t_a + \varnothing_p\right) \in [-\pi, 0); \\ r - \left(R_{BI}(0) + 2\cos(\beta/2)r_p\sqrt{1 - \left(\frac{f_a\lambda}{2\cos(\beta/2)r_p\omega}\right)^2}\right), \\ \qquad\qquad \left(\omega t_a + \varnothing_p\right) \in [0, \pi]; \end{cases} \tag{11}$$

From Equation (11), the energy trajectory of scatterers is related to the rotating radius $r_p$ and Doppler frequency $f_a$. It is worth pointing out that the energy trajectories of scatterers with the same rotating radius $r_p$ in the RD domain are consistent. Furthermore, the greater the rotating radius $r_p$ is, the larger the span of RCM for the energy envelope becomes. As a result, based on the symmetric property of the energy trajectory in the RD domain, the span of the energy trajectory distributed at $\left(\omega t_a + \varnothing_p\right) \in [0, \pi]$ is analyzed to illustrate the RCM. Conducting the derivative of $\mathcal{R}\left(r_p, f_a\right)$ with respect to $f_a$ at $\left(\omega t_a + \varnothing_p\right) \in [0, \pi]$ and setting it to be zero obtains

$$\frac{d\left[R_{BI}(0) + 2\cos(\beta/2)r_p\sqrt{1 - \left(\frac{f_a\lambda}{2\cos(\beta/2)r_p\omega}\right)^2}\right]}{df_a} = 0 \frac{f_a\lambda^2}{\sqrt{\left(2r_p\omega\cos(\beta/2)\right)^2 - (f_a\lambda)^2}} = 0 \tag{12}$$

The maximal value of $\mathcal{R}\left(r_p, f_a\right)$ is obtained under the condition of $f_{amax} = 0$, and it is

$$\mathcal{R}\left(r_p, f_{amax}\right) = R_{BI}(0) + 2\cos(\beta/2)r_p \tag{13}$$

Meanwhile, the value of $f_a$ should not exceed half of the Doppler bandwidth. Therefore, the minimum value is based on the condition of $f_{amin} = \pm\cos(\beta/2)r_p\omega/\lambda$, given by

$$\mathcal{R}\left(r_p, f_{amin}\right) = R_{BI}(0) + \sqrt{3}\cos(\beta/2)r_p \tag{14}$$

Therefore, from Equations (13) and (14), the span $\Delta\delta$ of the envelope trajectory in the RD domain is

$$\Delta\delta = \left|\mathcal{R}\left(r_p, f_{amax}\right) - \mathcal{R}\left(r_p, f_{amin}\right)\right| = \left|2\cos(\beta/2)r_p - \sqrt{3}\cos(\beta/2)r_p\right| \tag{15}$$

It should be pointed out that, from Equation (15), the span $\Delta\delta$ is increased with the increase in the rotating radius $r_p$, which would have an excess of one range resolution $c/(2B_r\cos(\beta/2))$ [33]. This would prevent the focus of the scatterer. As a result, the large RCM should be corrected into a range cell so as to produce a well-focused image.

By introducing new variables and substituting them into Equation (11), one obtains the discrete envelope trajectory of the scatterer as

$$\mathcal{R}(m, n) = \begin{cases} m - \left(R_{BI}(0) - 2\cos(\beta/2)m_p(c/f_s)\sqrt{1 - \left(\frac{n\lambda \cdot PRF}{2N\cos(\beta/2)m_p(c/f_s)\omega}\right)^2}\right), \\ \left(\frac{\omega n}{PRF} + \varnothing_p\right) \in [-\pi, 0); \\ m - \left(R_{BI}(0) + 2\cos(\beta/2)m_p(c/f_s)\sqrt{1 - \left(\frac{n\lambda \cdot PRF}{2N\cos(\beta/2)m_p(c/f_s)\omega}\right)^2}\right), \\ \left(\frac{\omega n}{PRF} + \varnothing_p\right) \in [0, \pi]; \end{cases} \tag{16}$$

where $M$ and $N$ denote the sizes of signals in the RD domain, $r = m \cdot (c/(2f_s))$, $r_p = m_p \cdot (c/(2f_s))$, $m = 1, 2, 3 \cdots M$, $n$ is an integer, and $n \in [-N/2, N/2]$, and $f_a = n \cdot (PRF/N)$.

For every Doppler channel $n$ in Equation (16), we circularly shift the data with ROUND[·]. (ROUND[·] denotes the circular shift operator. When ROUND[$\mathcal{R}(m,n)$] > 0, the data is circularly shifted upward. Otherwise, the data is shifted downward), and then map the data into the corresponding locations. Figure 3 depicts the data mapping diagram via the circular shift operation, where the data $r_p^1$. in Figure 3a is mapped to $r_p^{m-v+1}$ in the Doppler channel $n$ that, shown in Figure 3b.

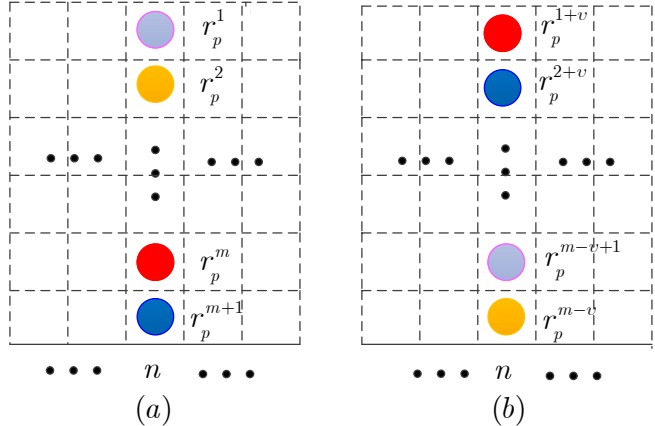

**Figure 3.** Mapping of data with circular shift operation. where the data $r_p^1$ in (**a**) is mapped to $r_p^{m-v+1}$ in the Doppler channel n that, shown in (**b**).

In this paper, the circular shift operation expression is denoted by

$$\Re(m,n) = \text{ROUND}(\mathcal{R}(m,n)) \tag{17}$$

where $\Re(m,n)$ denotes the result of the circular shift operation for the curve trajectory $\mathcal{R}(m,n)$, and the diagram is shown in Figure 2c1, in which the red solid line is the original trajectory and the red dotted line is the result after circular shift operation.

After applying the circular shift operation in Equation (17) to all Doppler channels, the corresponding analytical expression in the RD domain can be described as follows

$$sS(r, f_a) = \begin{cases} \sigma_p w(f_a/B_a) \cdot sinc\left[\frac{\pi B_r}{c}\left(r - \left(R_{BI}(0) - 2cos(\beta/2)r_p\right)\right)\right] \\ \times exp\left(j2\pi f_a \frac{\varnothing_p}{\omega}\right) exp\left(j\varphi_1(r_p, f_a)\right), \left(\omega t_a + \varnothing_p\right) \in [-\pi, 0); \\ \sigma_p w(f_a/B_a) \cdot sinc\left[\frac{\pi B_r}{c}\left(r - \left(R_{BI}(0) - 2cos(\beta/2)r_p\right)\right)\right] \\ \times exp\left(j2\pi f_a \frac{\varnothing_p}{\omega}\right) exp\left(j\varphi_2(r_p, f_a)\right), \left(\omega t_a + \varnothing_p\right) \in [0, \pi]; \end{cases} \tag{18}$$

It is worth recalling that, after finishing the circular shift operation, the energies of scatterers with curve trajectory distribution are now distributed into one range cell, shown in Figure 2c1. Consequently, the large RCM is corrected by using the circular shift operation. In order to produce well-focused ISAR images, the high order phase terms that contains nonstationary DFM should be compensated. Therefore, the compensation function $H(r_p, f_a)$, in this work, can be designed as

$$H(r_p, f_a) = \begin{cases} exp\left(-j\varphi_1(r_p, f_a)\right), \left(\omega t_a + \varnothing_p\right) \in [-\pi, 0); \\ exp\left(-j\varphi_2(r_p, f_a)\right), \left(\omega t_a + \varnothing_p\right) \in [0, \pi]; \end{cases} \tag{19}$$

Multiplying Equation (19) with Equation (18) yields

$$
S(r, f_a) = \begin{cases}
\sigma_p w(f_a/B_a)\cdot\text{sinc}\left[\frac{\pi B_r}{c}\left(r - \left(R_{BI}(0) - 2\cos(\beta/2)r_p\right)\right)\right] \\
\quad \times exp\left(j2\pi f_a \frac{\varnothing_p}{\omega}\right), \left(\omega t_a + \varnothing_p\right) \in [-\pi, 0); \\
\sigma_p w(f_a/B_a)\cdot\text{sinc}\left[\frac{\pi B_r}{c}\left(r - \left(R_{BI}(0) - 2\cos(\beta/2)r_p\right)\right)\right] \\
\quad \times exp\left(j2\pi f_a \frac{\varnothing_p}{\omega}\right), \left(\omega t_a + \varnothing_p\right) \in [0, \pi];
\end{cases}
\tag{20}
$$

From Equation (20), the energies at $\left(\omega t_a + \varnothing_p\right) \in [-\pi, 0)$ and $\left(\omega t_a + \varnothing_p\right) \in [0, \pi]$ are symmetric about $r = R_{BI}(0)$, which can be used to transform the energies of those two parts into the same range cell, shown by the red dotted line in Figure 2d1. After that, the analytical expression of the signal is

$$
sS_{comp}(r, f_a) = \sigma_p w(f_a/B_a)\cdot\text{sinc}\left[\frac{\pi B_r}{c}\left(r - \left(R_{BI}(0) + 2\cos(\beta/2)r_p\right)\right)\right] \times exp\left(j2\pi f_a \frac{\varnothing_p}{\omega}\right)
\tag{21}
$$

By performing inverse Fourier transform (IFT) to $sS_{comp}(r, f_a)$, the analytical expression can be expressed as

$$
\begin{aligned}
ss_{comp}(r, t_a) &= F_{f_a}^{-1}\left[sS_{comp}(r, f_a)\right] \\
&= \sigma_p\cdot\text{sinc}\left[\frac{\pi B_r}{c}\left(r - \left(R_{BI}(0) + 2\cos(\beta/2)r_p\right)\right)\right] \times \text{sinc}\left[\frac{\pi B_a}{\omega}\left(t_a - \varnothing_p\right)\right]
\end{aligned}
\tag{22}
$$

where $F_{f_a}^{-1}[\cdot]$ denotes the IFT operator along the Doppler frequency $f_a$. Based on the analytical expression from Equation (22), the coordinate of scatterer $P$ is determined by $\left(R_{BI}(0) + 2\cos(\beta/2)r_p, \varnothing_p\right)$. Furthermore, to suppress the high sidelobe of the scatterer, the signal energies below half of the maximum energy are restrained.

However, in reality, the targets comprise multiple scatterers with different rotating radii and initial phases. According to the analysis mentioned above, the energy of scatterers with the same rotating radius can be corrected into one range cell by use of the same circular shift operation. However, for the one with different rotating radii, the energy corresponding to the circular shift operation is concentrated into the identical range unit, but the RCM of others also exist. The diagram of which is described in Figure 2c1, where the red solid line with curve trajectory is transformed into the red dotted line that is concentrated into a range cell. However, the blue solid line is transformed into the blue dotted line that is also a curve trajectory.

To solve this issue, the following steps are designed to correct the RCMs for all scatterers with different rotating radii, and the realization procedure of which is given as follows.

Step (1) Applying Equatioin (17) to calculate the energy of the scatterer with the rotating radius $r_{pi}$ to concentrate the corresponding scatterer energy into one range unit. Meanwhile, the corresponding high order phase can be compensated, and the well-focused scatterers with rotating radius $r_{pi}$ are produced via utilizing IFT in the azimuth dimension as a subimage $I_i$, which is demonstrated from Figure 2c1 to Figure 2e1.

Step (2) Repeat Step (1) for other scatterers with rotating radius $r_{p(i+1)}$ until the rotating radius $r_{max}$ is found, and obtain all the subimages $I_i$, which are indicated from Figure 2c2 to Figure 2e2.

Step (3) The bistatic ISAR images of non-cooperative targets with rapid spinning motions are reconstructed via merging the subimage $I_i$ into an image presented in Figure 2e1,2e2 to Figure 2f.

Finally, the whole flowchart of bistatic ISAR imaging for rapidly spinning space non-cooperative targets is shown in Figure 4.

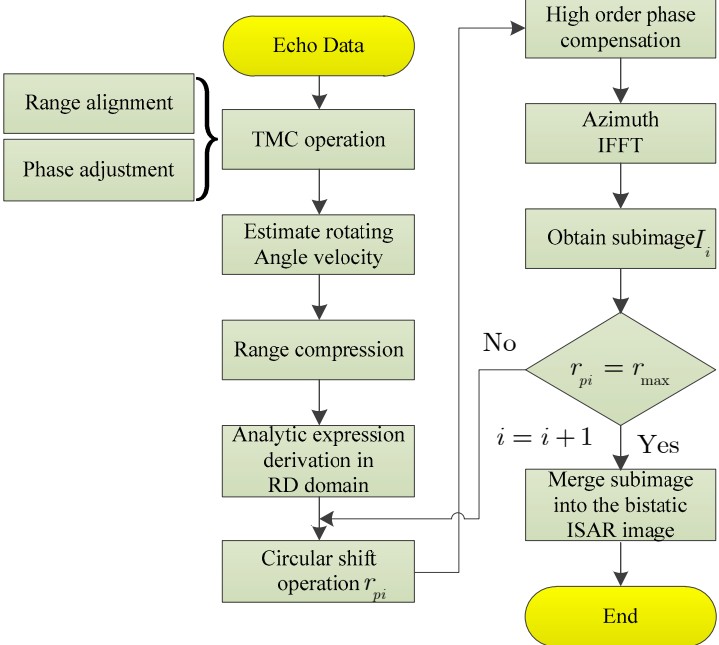

**Figure 4.** The whole flowchart for proposed bistatic ISAR imaging method.

### 2.3. Computational Load Analysis

In this section, the computational load for our proposed algorithm is analyzed quantitatively. For comparison purposes, the GRT method in [30] is introduced in this work. Generally, the computational load for an $N$-point FT or IFT is $N\log_2(N)$ floating-point operations (FLOPs), and $N$ FLOPs for the one-time complex multiplication of $N$. -point data. Now, we suppose the values of the range dimensional and azimuth dimensional are $N_r$ and $N_a$, respectively. $M$, $P$ and $Q$. denote the sub-image imaging within our method, the number for searching rotating radii and the initial phase using the GRT method, respectively.

For the GRT method, the imaging procedure mainly includes performing range compression and searching parameters for the rotating radius and initial phase. Thus, the computational complexity of range compression is $O\left[2N_aN_r\log_2(N_r)+N_aN_r\right]$. and the computational complexity for energy accumulation along with different trajectories is $O\left[2N_aN_r\log_2(N_r)+N_aN_r\right]$. In brief, the total computational complexity of the GRT algorithms is

$$C_{GRT}=O\left[2N_aN_r\log_2(N_r)+N_aN_r+PQN_a\right] \tag{23}$$

Based on the whole procedure for bistatic ISAR imaging in Figure 4, the proposed method is made up of one-time range compression and $M$-time sub-image ISAR merging. As a result, the total computational load for the proposed algorithm is

$$C_{proposed}=O\left[2N_aN_r\log_2(N_r)+N_aN_r+N_rN_a\log_2(N_a)+M\left(N_aN_r+N_rN_a\log_2(2N_a)\right)\right] \tag{24}$$

In conclusion, from Equations (23) and (24), although our proposed method repeats multiple ISAR imaging, the search for different rotating radii and initial phases with GRT imaging is actually more time-consuming.

## 3. Simulation Results and Analysis

In this section, several simulation experiments are performed to verify the validity of the presented approach, and the simulation parameters for bistatic ISAR imaging are provided in Table 1.

**Table 1.** Simulation parameters of bistatic ISAR imaging.

| | |
|---|---|
| Carrier frequency | 11 GHz |
| Frequency bandwidth | 1.5 GHz |
| Pulse width | 1 us |
| PRF | 2000 Hz |
| Position of Tx | [0, 13.05 km] |
| Position of Rx | [0, −13.05 km] |
| Baseline of Tx and Rx | 26.1 km |
| Target imaging center | [99.14 km, 0] |
| Rotating angle velocity | 6.28 rad/s |
| Bistatic angle | pi/12 |

### 3.1. Bistatic ISAR Imaging for Single Scatterer

Utilizing the parameters listed in Table 1, the bistatic ISAR results for single scatterers with polar coordinates $r_p = 1.3$, $\varnothing_p = \frac{\pi}{4}$ are depicted in Figure 5. The single scatterer model and profile of echoes after range compression are distributed in Figure 5a,b. It can be noted that the trajectory of the profile is a form of trigonometric function that spans multiple range cells. By using the GRT method, the energies of scatterers are accumulated along the trigonometric trajectory. Figure 5c provides the signal envelope in the RD domain. It is noticeable the energies are also still spreading across several range cells. The circular shift operation results of the energies distributed at $\left(\omega t_a + \varnothing_p\right) \in [-\pi, 0)$ and $\left(\omega t_a + \varnothing_p\right) \in [0, \pi]$ are provided in Figure 5d,f, respectively. At the same time, the corresponding results after high-order phase compensation are provided in Figure 5e,g. It should be pointed out that, with the circular shift operation, the energies of scatterers with large RCM have been corrected into a range cell. Figure 5h,i show the ISAR image utilizing the GRT method and our proposed method, respectively. From Figure 5h,i, the sidelobe imaging result obtained by the GRT method is higher than that of our proposed method, which would affect the main lobe of weak scatterers. To conclude, all of this clearly shows that our proposed approach has superior imaging results to those of the GRT method.

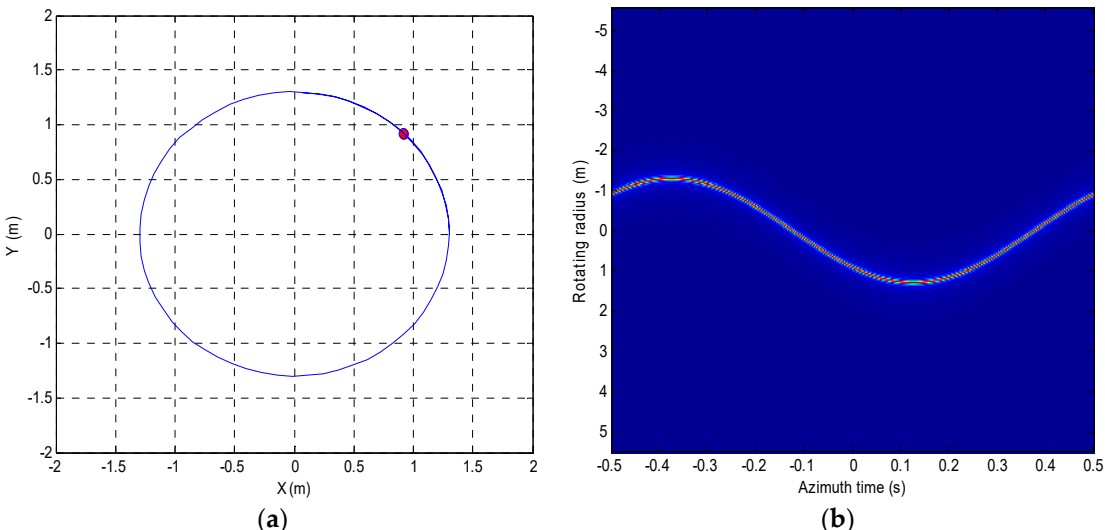

(a)

(b)

**Figure 5.** *Cont.*

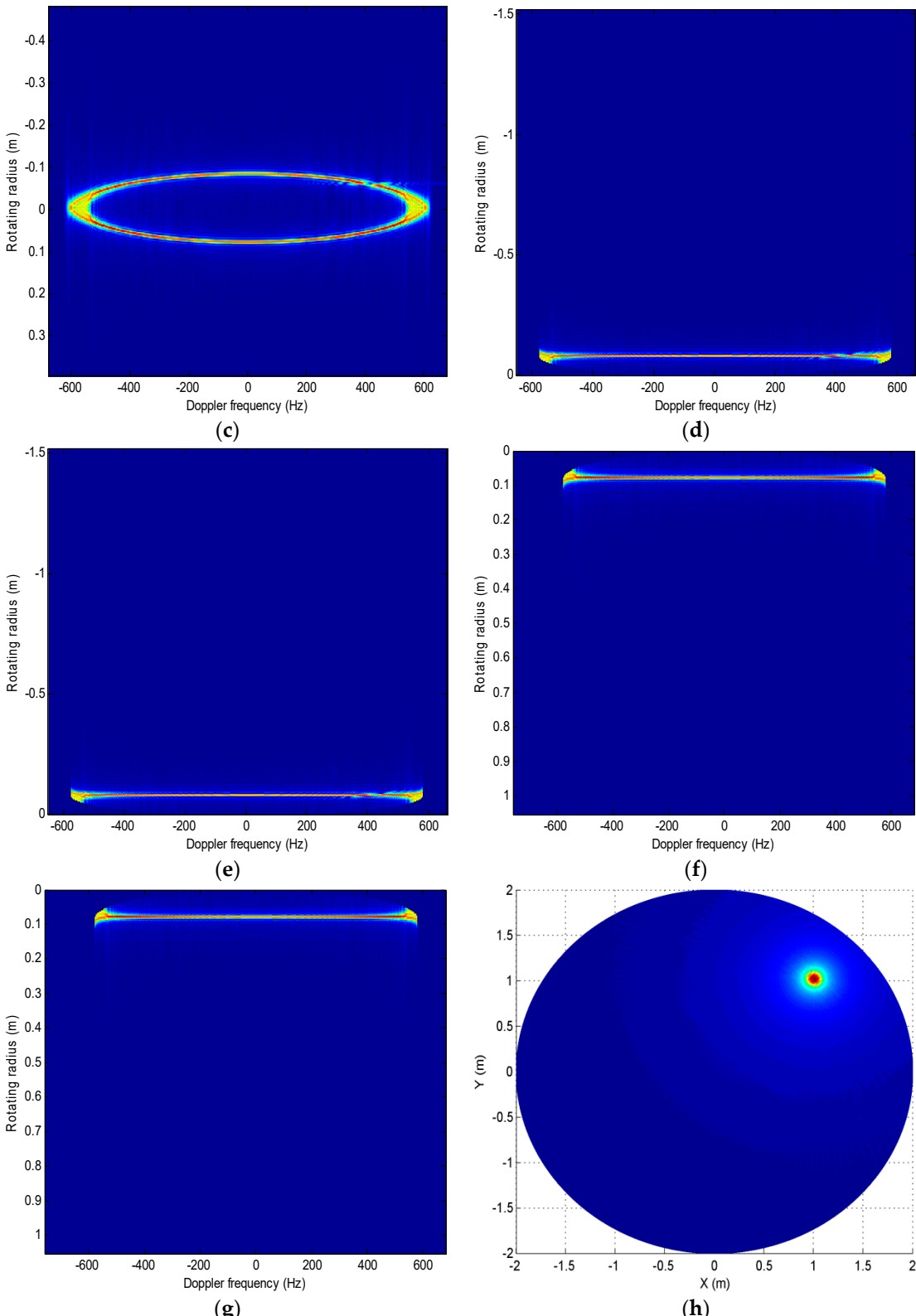

**Figure 5.** *Cont.*

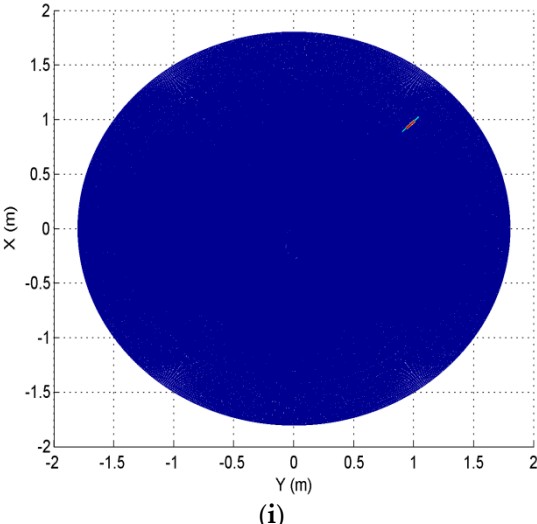

**(i)**

**Figure 5.** Single scatterer imaging with proposed method. (**a**) Single scatterer model distribution. (**b**) The profile of echoes after range compression. (**c**) The results in range-Doppler (RD) domain. (**d**) The circular shifting operation result at $\left(\omega t_a + \varnothing_p\right) \in [-\pi, 0)$. (**e**) The result of high order term phase compensation at $\left(\omega t_a + \varnothing_p\right) \in [-\pi, 0)$. (**f**) The circular shifting operation result at $\left(\omega t_a + \varnothing_p\right) \in [0, \pi]$. (**g**) The result of high order phase term compensation at $\left(\omega t_a + \varnothing_p\right) \in [0, \pi]$. (**h**) The imaging result with GRT method. (**i**) The imaging result with our proposed method.

### 3.2. Imaging of Multiple Scatterers

The scatterer's distribution is provided in Figure 6a, and the polar coordinates of scatterers are provided in Table 2. It is clear from Figure 6b that the profiles of echoes with the forms of trigonometric functions still span multiple range cells. Figure 6c provides the two-dimensional (2D) imaging result using the GRT method. Figure 6d shows the 2D imaging results of our proposed method. The estimated coordinates and errors with GRT and our proposed method are listed in Table 3, where the upper row and bottom row, respectively, indicate the estimated coordinates and the errors. It should be pointed out that the imaging performance of our method is superior to that of the GRT method. In addition, the evaluation indicator of the image contrast [34] is provided to quantify the imaging result. It represents the ratio of target to background brightness in an image. The greater the image contrast is, the clearer the image will become. Thus, it is more beneficial to extract and classify the objects. The image contrast is defined by

$$C = \frac{\sqrt{E\left((I(m,n) - E(I(m,n)))^2\right)}}{E(I(m,n))} \tag{25}$$

where $I(m,n)$ and $E(I)$ represent the amplitudes of the image coordinate $(m,n)$ and mean value of the $I$.

**Table 2.** The coordinates of multiple scatterers.

| | | | |
|---|---|---|---|
| (0.5, −2.3562) | (0.5, −0.7854) | (0.5, 0.7854) | (0.5, 2.3562) |
| (1.1, −2.3562) | (1.1, −0.7854) | (1.1, 0.7854) | (1.1, 2.3562) |
| (1.5, −2.3562) | (1.5, −0.7854) | (1.5, 0.7854) | (1.5, 2.3562) |

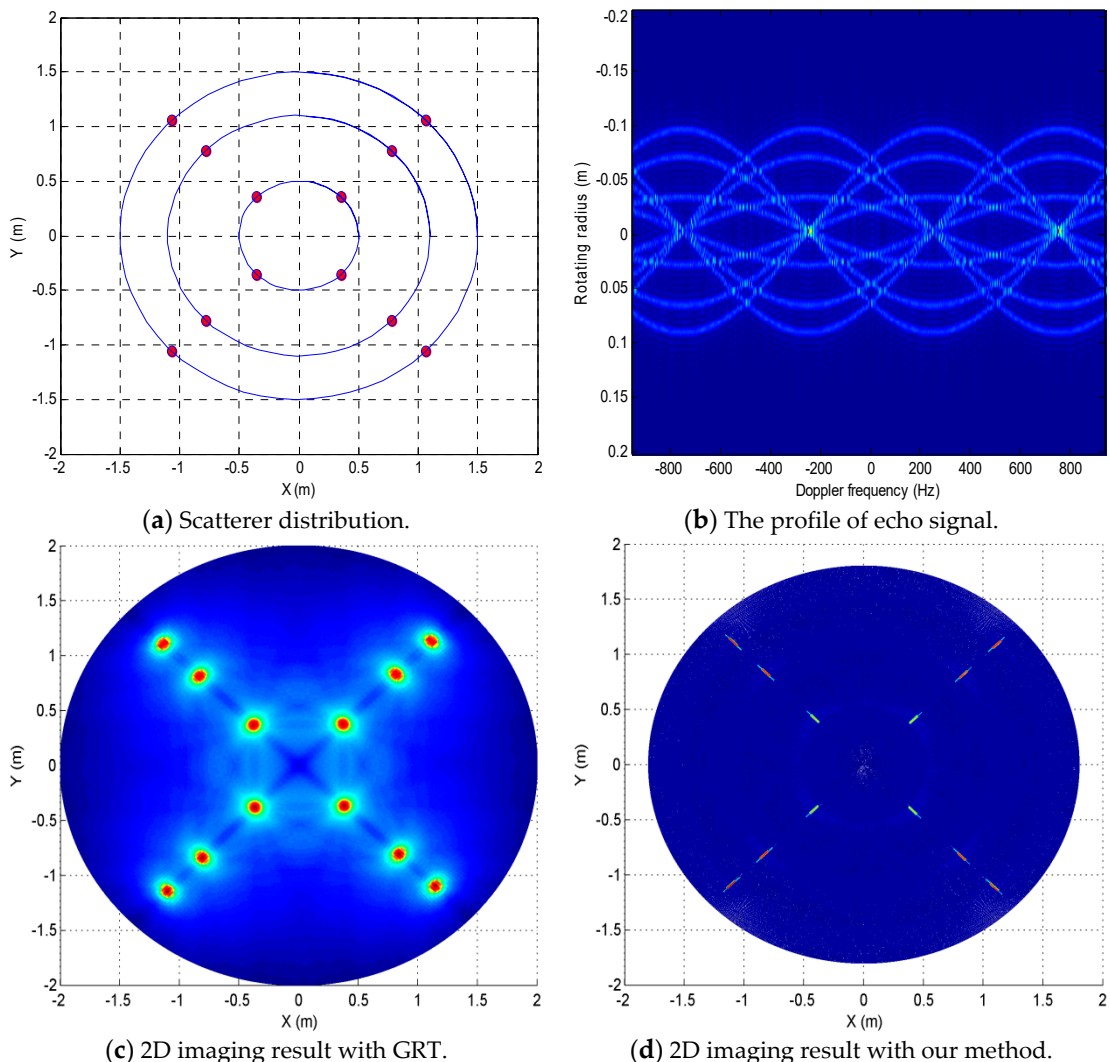

(**a**) Scatterer distribution.

(**b**) The profile of echo signal.

(**c**) 2D imaging result with GRT.

(**d**) 2D imaging result with our method.

**Figure 6.** Scatterer distribution of the simulated model and the imaging result. (**a**) Multiple scatterer model distribution. (**b**) The profile of echoes after range compression. (**c**) Imaging result with GRT method. (**d**) Bistatic ISAR Imaging result with the proposed method.

**Table 3.** Error between the estimated coordinates and the errors with generalized radon transform (GRT).

| GRT | Our Proposed Method | GRT | Our Proposed Method |
|---|---|---|---|
| (0.5293, −2.3752) | (0.5119, −2.366) | (0.5287, −0.8044) | (0.5284, −0.7948) |
| (0.0293, 0.019) | (0.0119, 0.0098) | (0.0287, 0.019) | (0.0284, 0.0094) |
| (0.53, 0.7664) | (0.5284, 0.776) | (0.5313, 2.3371) | (0.5284, 2.347) |
| (0.03, 0.019) | (0.0284, 0.0094) | (0.0313, 0.0191) | (0.0284, 0.0092) |
| (1.1653, −2.3752) | (1.106, −2.366) | (1.162, −0.8044) | (1.106, −0.7948) |
| (0.0653, 0.019) | (0.006, 0.0098) | (0.062, 0.019) | (0.006, 0.0094) |
| (1.162, 0.7664) | (1.106, 0.776) | (1.164, 2.3371) | (1.106, 2.347) |
| (0.062, 0.019) | (0.006, 0.0094) | (0.064, 0.0191) | (0.006, 0.0092) |
| (1.5867, −2.3752) | (1.503, −2.366) | (1.586, −0.8044) | (1.503, −0.7948) |
| (0.0867, 0.019) | (0.003, 0.0098) | (0.086, 0.019) | (0.003, 0.0094) |
| (1.5847, 0.7664) | (1.503, 0.776) | (1.5847, 2.3371) | (1.503, 2.347) |
| (0.0847, 0.019) | (0.003, 0.094) | (0.0847, 0.0191) | (0.003, 0.0094) |

The average results of 100 trials for image contrast are collected, with SNR ranging from −10 dB to 20 dB in steps of 1 dB, to numerically compare the performance of two methods. The curves of image contrast against SNRs obtained by two methods are shown in Figure 7. It should be pointed out that

the image contrast increases with the increase in SNR. It is worth noting that our proposed method always achieves a larger image contrast than the GRT method.

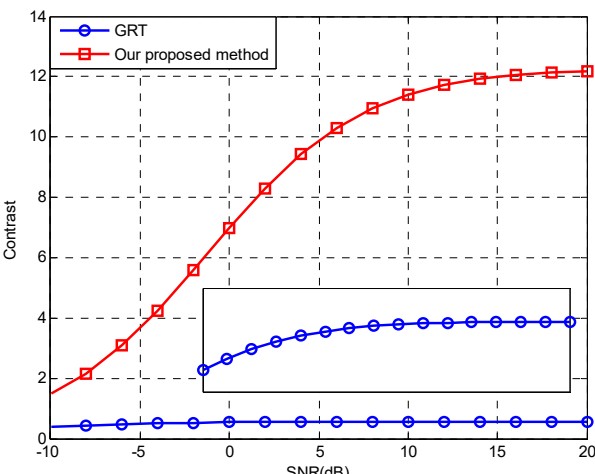

**Figure 7.** Cures of image contrast versus different signal to noise ratios (SNRs).

### 3.3. Estimation Error Analysis with Different SNRs

In this section, the anti-noise performance is analyzed by adding white Gaussian noise to the echo data (after range compression). Figure 8a,b indicate the reconstruction errors of scatterers against different SNRs. Considering the imaging principle of GRT for searching different rotating radii and initial phases, the tradeoff between fixed search step size and imaging efficiency should be noted. Therefore, a reasonable search step is taken for GRT imaging. It should be noted that the reconstruction veracity for the rotating radius and initial phases is improved as SNR increases, and the superiority of our introduced method in contrast to the GRT method is more conspicuous thanks to the coherent accumulations along the azimuth dimension. In conclusion, the proposed method presents a higher reconstruction precision for the rotating radius and initial phases under a low SNR circumstance.

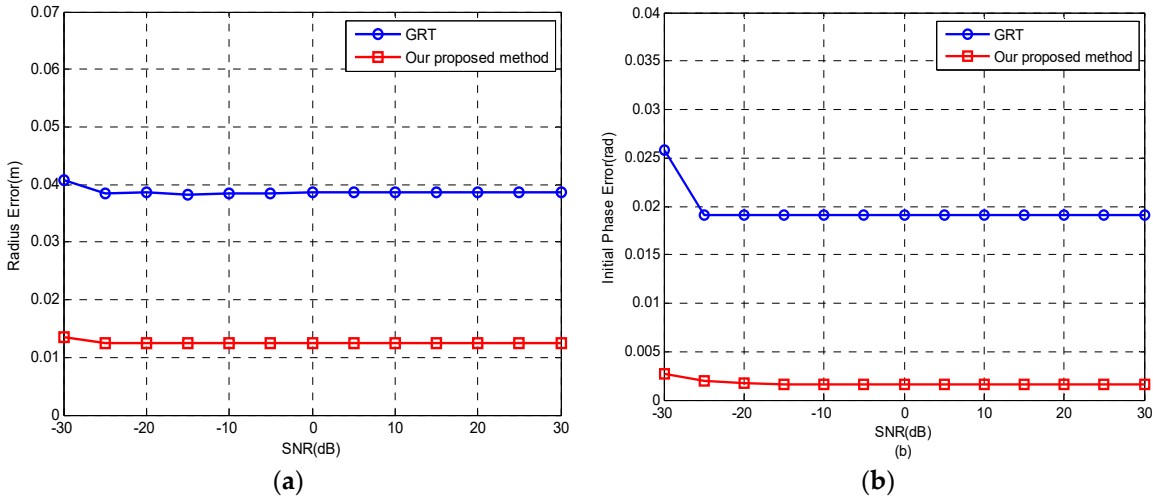

**Figure 8.** Estimation Error of rotates radius and initial phase. (**a**) Rotating radius. (**b**) Initial phase.

### 3.4. Electromagnetic Modeling Simulation

In this section, simulations with electromagnetic (EM) data are performed to verify the performance for the proposed approach. 'FEKO' software is utilized to create the radar cross-section (RCS) data. Noteworthily, predicting these RCS data would be a resourceful and economical way to acquire the

echoes for targets with rapid spinning because measuring real-world data has practical difficulties. The well-known physical optical (PO) [35] technique is utilized, which is one of the most widely adopted techniques for high-frequency EM computation. The parameters of the EM model are provided in Table 4. Due to limited laboratory equipment, an interpolation operation is conducted to improve the numbers for frequencies and pulses. Figure 9a provides the computer-aided design (CAD) model. It should be observed that the CAD model is a cone target, the parameters of which are provided in Table 4. The imaging results of the EM data are provided in Figure 9b. In conclusion, the results indicate that the proposed approach has the ability to image rapidly spinning targets.

**Table 4.** Parameters of electromagnetic (EM) model.

| | | | |
|---|---|---|---|
| Start frequency | 6.5 GHz | Carrier frequency | 7 GHz |
| End frequency | 7.5 GHz | Number of pulses | 2001 |
| Number of frequency | 6200 | Base radius | 3 m |
| Pitch angle | 60° | Height | 1.5 m |
| Bistatic angle | Pi/4 | Rotating velocity | 6.28 rad/s |
| Pulse Width | 1 us | PRF | 2000 |

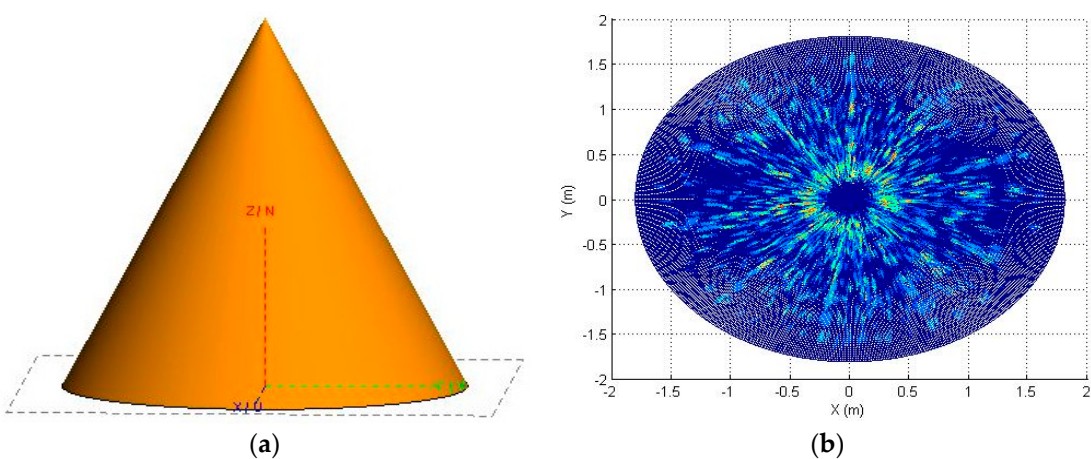

(**a**)   (**b**)

**Figure 9.** EM model of target in 'FEKO' software. (**a**) Computer-aided design (CAD) model. (**b**) Imaging result with our proposed approach.

*3.5. Comparison for Computational Load*

In this section, the computation load for the GRT approach and the presented algorithm are compared with each other; the running times via utilizing different scene sizes are presented in Figure 10, which are obtained by utilizing a computer device with an Intel(R) Core (TM), i5-8400, CPU clocked frequency at 2.80 GHz, memory 8 GB, Windows 10 operating system, and MATLAB version 2014a. It is noteworthy that the running time for our presented method is less than that of the GRT method, which is in agreement with the analysis mentioned above. In addition, with an increase in the image scene size, the superiority of our proposed method in regards to implementation time becomes more evident.

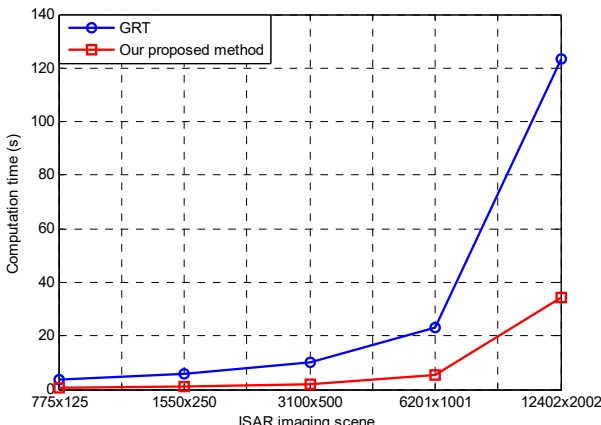

**Figure 10.** The comparison of computational complexity.

## 4. Discussion

Inverse synthetic aperture radar (ISAR) plays a crucial role in the detection, recognition or identification of rapidly spinning targets, because electromagnetic images of non-cooperative targets with rapid rotating motions can be obtained. Conventional studies of ISAR imaging have focused mainly on monostatic radar configurations in which a transmitter and a receiver are collocated. However, considering the imaging problem for stealthy targets, for instance, the monostatic ISAR technique suffers from restrictions. While utilizing a monostatic radar configuration, the electromagnetic energy is reflected by the stealthy targets to the direction other than that of the receiver line of sight (LOS), which yields a decrease in the SNR for a received radar echo signal. Compared with monostatic radar configuration, the bistatic radar configuration has many advantages in detection scope, concealment, anti-interference capability, and so on. Therefore, the bistatic ISAR configuration is utilized in imaging for targets with rapid spinning motions.

The main problem of imaging for targets with rapidly spinning motion is that the rotating angular is larger than 360 degree. Thus, the instantaneous slant range contains a trigonometric function that cannot be expanded via Taylor's series. Therefore, the large RCM and nonstationary DFM of echoes in the coherent processing interval restrain the well-focused ISAR image. Meanwhile, the low noise affects the imaging for targets with rapidly spinning motion. To overcome the obstacles mentioned above, an effective bistatic ISAR imaging approach with circular shift operation in the RD domain is proposed based on the azimuth spatial invariance. The large RCM is corrected by using a circular shift operation, and the nonstationary DFM can also be compensated along the rotating radius direction. At the same time, the proposed approach has a better robustness to noise, as shown in Figure 8. When SNR is at the level of −30 dB to 30 dB, the reconstructed error for the rotating radius and initial phases is small. The real-time performance of our proposed method is also high in contrast to that of the GRT method, as provided in Figure 10. The result is especially evident when the imaging scene size is $12,402 \times 2002$. We also conducted simulations with electromagnetic data, which is a resourceful and economical way to acquire the echoes of targets with rapid spinning motions because measuring real-world data has practical difficulties.

Though well-focused two-dimensional ISAR images for targets with rapid spinning motions can be obtained, the three dimensional ISAR images that provide more information are not explored in this work. In conclusion, three-dimensional ISAR imaging research for targets with rapid spinning motions can be conducted in future works.

## 5. Conclusions

In the case of the rapidly spinning targets in bistatic ISAR, the large RCM and nonstationary DFM of echo signals in coherent processing intervals restrain well-focused bistatic ISAR imaging, which creates difficulties in the recognition of targets. To overcome those obstacles, in this work,

a high-efficiency bistatic ISAR imaging approach with circular shift operation in the RD domain is proposed based on the azimuth spatial invariance. Firstly, the echoes of targets are transformed into the RD domain, and the accurate analytical derivation is derived by utilizing the theory of POSP. Based on that, secondly, the envelope trajectories of RCM are corrected by efficient circular shift operations, and the time-varying DFM can also be compensated along the rotating radius direction. Finally, several simulations are implemented to show the availability of the proposed algorithm compared with existing approaches. Meanwhile, simulation results utilizing bistatic RCS data computed with the PO technique are also presented to confirm the usability for the proposed method. In conclusion, the proposed approach provides a good tradeoff between the performance and computation time in obtaining clear images for non-cooperative targets with rapid spinning motions in a noisy environment.

**Author Contributions:** Z.Y., X.T. proposed the method, designed the experiments, and conceived and analyzed the data; Z.Y. and D.L. performed the experiments and wrote the paper; and H.L. and G.L. revised the paper. All authors have read and agreed to the published version of the manuscript.

**Funding:** This work was supported, in part, by the National Natural Science Foundation of China (grant No. 61971075), in part by the graduate research and innovation foundation of Chongqing, China (grant No. CYB19058, grant No. CYB18068), in part by The Key Project of Application and Development of Chongqing (cstc2019jscx-fxyd0354), in part by the Chongqing Research Program of Basic Research and Frontier Technology (grant no. cstc2018jcyjAX0351), in part by the Pre-research Fund Project (61404130114 and 61404130219), in part by the Fundamental Research Funds for the Central Universities (grant No. 2019CDQYTX012), in part by the National Natural Science Foundation of China (grant No. 51877015), and in part by the Joint Research Fund in Astronomy through a Cooperative Agreement with the National Natural Science Foundation of China (NSFC) (grant No. U1831117).

**Acknowledgments:** The authors would like to thank all the anonymous reviewers and editors for their useful comments and suggestions that greatly improved this paper.

**Conflicts of Interest:** The authors declare no conflict of interest.

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
