# Peer review of "A Fast Bistatic ISAR Imaging Approach for Rapidly Spinning Targets via Exploiting SAR Technique"

_remotesensing, doi:10.3390/rs12132077_

Round 1

Reviewer 1 Report

The authors have greatly improved the first version of their manuscript.
Nevertheless, some errors remain, for example :
- The relationship that defines contrast is wrong. It is the calculation of the RMS contrast of an image. Therefore, in expression (25) instead of putting I^2, one should put I.
- Correct a typo error in the x-axis of figure 10: instead of ISAR imageing scence, put ISAR imaging scene.

Translated with www.DeepL.com/Translator (free version)

Reviewer 2 Report

The Comments and Suggestions are not reflected enough to be reconsidered for publication in this journal.

Authors did not include proper answers to previous comments of Reviewer-2:

  1. Read Measurement: no answer.
  2. Comment (4) comparing IRF (Impulse Response Function): authors included image contrast comparison rather than IRF for performance evaluation.
  3. Comment (5) CAD models or real target: authors used eight balls rather than CAD models, nor real target.

Reviewer 3 Report

This paper is a re-submitted version of an earlier paper. The authors have addressed all of my comments and suggestions. It is acceptable now.

Author Response

Thanks for reviewer’s time and efforts spent to review this manuscript.

Round 2

Reviewer 2 Report

The author answered all the comments for publication.

This manuscript is a resubmission of an earlier submission. The following is a list of the peer review reports and author responses from that submission.

Round 1

Reviewer 1 Report

This study deals with a very important theme in the real world: Bistatic ISAR imaging.
The first criticisms for the authors are that the work presented is based on simulations whereas in the real world, the effect of noise disrupts the imaging process.
It is therefore important that the authors note in the introduction:  "In  real world, ISAR imaging must use signal processing methods to characterize signals of interest [1]. "

[1] FEMMAM, N. K.M’SIRDI, and A. OUAHABI, “ Perception and characterization of materials using signal processing techniques ”, IEEE Transactions on Instrumentation & Measurements, Vol. 50, N°5, pp. 1203-1211, Oct. 2001.

"A simple way to eliminate or at least reduce noise is to use a wavelet approach[2-3]."

Some notations, in particular the expression (17), are poorly chosen since R is generally reserved for the set of real numbers.
The result of the inverse Fourier transform (IFT) given in (22) seems to me to be wrong.

Figure 7 lacks justification and clarity.
Figure 8 is unnecessary since the computation time depends on the machine used and should not be presented as a performance argument.
In addition, obtaining the measurement points shown in  figure 8 is not convincing.

Reviewer 2 Report

Modeling and Simulation with ideal point targets is not enough to be considered for publication in this journal.

More serious M&S and real measurements should be followed with this study for publication.

I should recommend the authors to submit to a Letter rather than Full Paper (such as Remote Sensing) due to the following reasons.  

1. The title and the contents (scope) of this study do not match.    

The study carried out computer simulations with ideal point targets but did not carried our measurements while readers expect general targets as a scope of this article from the title.    

Authors should have named the title from what they did.  

2. The Conclusions are too weak.    

It is because the contents of this study is not enough for a full paper.    

I suggest to shorten the length of this manuscript, for example introduction, expressions (pass to Appendix), and submit to a Letter.  

3. The good point of this manuscript is details of the imaging method in Section 2.2.  

4. To prove the proposed method, authors should have shown (proven) the performance of the method for targets.    

It is general method to prove the novelty of the new algorithm for point target with comparing IRF (Impulse Response Function) with conventional methods.    

Readers would have difficulty to recognize the performance the new method without typical performance evaluation.  

5. Target means not only ideal point targets.    

It should be identified for readers not to be confused.    

Author should have carried out modeling and simulation for various targets (for examples, CAD models), or real measurements for real targets.    

If not, hope to rename the title.

Reviewer 3 Report

Please see the attached report.
